# CO electrolysis to multicarbon products over grain boundary-rich Cu nanoparticles in membrane electrode assembly electrolyzers

Hefei Li[1,2,4], Pengfei Wei[1,4], Tianfu Liu[1,4], Mingrun Li[1], Chao Wang[1], Rongtan Li [1,2], Jinyu Ye [3], Zhi-You Zhou [3], Shi-Gang Sun [3], Qiang Fu [1], Dunfeng Gao [1] ✉, Guoxiong Wang [1] ✉ & Xinhe Bao [1]

Producing valuable chemicals like ethylene via catalytic carbon monoxide conversion is an important nonpetroleum route. Here we demonstrate an electrochemical route for highly efficient synthesis of multicarbon ($C_{2+}$) chemicals from CO. We achieve a $C_{2+}$ partial current density as high as $4.35 \pm 0.07$ A cm$^{-2}$ at a low cell voltage of $2.78 \pm 0.01$ V over a grain boundary-rich Cu nanoparticle catalyst in an alkaline membrane electrode assembly (MEA) electrolyzer, with a $C_{2+}$ Faradaic efficiency of $87 \pm 1\%$ and a CO conversion of $85 \pm 3\%$. Operando Raman spectroscopy and density functional theory calculations reveal that the grain boundaries of Cu nanoparticles facilitate CO adsorption and C – C coupling, thus rationalizing a qualitative trend between $C_{2+}$ production and grain boundary density. A scale-up demonstration using an electrolyzer stack with five 100 cm$^2$ MEAs achieves high $C_{2+}$ and ethylene formation rates of 118.9 mmol min$^{-1}$ and 1.2 L min$^{-1}$, respectively, at a total current of 400 A (4 A cm$^{-2}$) with a $C_{2+}$ Faradaic efficiency of 64%.

With the decline and depletion of oil resource, producing valuable chemicals like ethylene via the conversion of syngas, a mixture of CO and $H_2$ derived from coal, natural gas, and biomass, has been considered as an efficient nonpetroleum route[1]. In thermal catalysis, CO hydrogenation to ethylene proceeds with a stoichiometric $H_2$/CO ratio of 2. However, the $H_2$/CO ratio is usually less than 1 in the syngas prepared by coal gasification that is the most cost-effective way in syngas production[2]. This mismatch is addressed by water gas shift reaction which generates more $H_2$ at the expense of CO and produces $CO_2$. Moreover, while a high selectivity ~80% for light olefins among hydrocarbon products in CO hydrogenation can be achieved through oxide-zeolite (OX-ZEO) and Fischer-Tropsch synthesis (FTS) processes, 20 – 50% of the converted CO is transformed into $CO_2$ and methane[3–5]. The substantial $CO_2$ emission as well as the undesired methane

production results in a low carbon utilization efficiency in thermocatalytic CO hydrogenation. Therefore, there is an urgent need to develop more sustainable routes for CO conversion.

Electrocatalysis, when driven by renewable energy, provides an alternative route for catalytic conversion of important carbon resources such as CO. CO electrolysis, an electrocatalytic CO hydrogenation process at ambient temperature and pressure, utilizes water rather than $H_2$ as a hydrogen source, and electrochemically eliminates the formation of $CO_2$ by applying a negative potential on CO molecules (Supplementary Fig. 1). While high Faradaic efficiency (FE) towards preferred multicarbon ($C_{2+}$) products including ethylene, acetate, ethanol, and n-propanol has been reported[6,7], the practical application of CO electrolysis is still hindered by low current density and energy efficiency due to insufficient catalytic activity and large Ohmic

[1]State Key Laboratory of Catalysis, Dalian National Laboratory for Clean Energy, iChEM (Collaborative Innovation Center of Chemistry for Energy Materials), Dalian Institute of Chemical Physics, Chinese Academy of Sciences, Dalian 116023, China. [2]University of Chinese Academy of Sciences, Beijing 100049, China. [3]State Key Laboratory of Physical Chemistry of Solid Surfaces, iChEM, College of Chemistry and Chemical Engineering, Xiamen University, Xiamen 361005, China. [4]These authors contributed equally: Hefei Li, Pengfei Wei, Tianfu Liu. ✉e-mail: dfgao@dicp.ac.cn; wanggx@dicp.ac.cn

resistance in H-cells and flow cells[8–12]. In addition, a small portion of CO is converted to undesired methane as a by-product over some catalysts[13,14]. To address these challenges, herein, using a zero-gap alkaline membrane electrode assembly (MEA) electrolyzer, we achieve $CO_2$-free, high-rate synthesis of $C_{2+}$ products via CO electrolysis over a grain boundary (GB)-rich Cu nanoparticle catalyst, with a $C_{2+}$ partial current density of $4.35 \pm 0.07\,A\,cm^{-2}$ at a low cell voltage of $2.78 \pm 0.01\,V$. CO is exclusively converted to $C_{2+}$ products (-100% carbon selectivity) and its conversion reaches up to $85 \pm 3\%$ at a high converted CO rate of $65.1 \pm 2.3\,mL\,min^{-1}$. The presented performance for electrochemical synthesis of $C_{2+}$ chemicals is notable comparable to previously reported electrocatalytic and thermocatalytic CO hydrogenation processes. Operando Raman spectroscopy and density functional theory (DFT) calculations reveal that the GBs of Cu nanoparticles facilitate $C - C$ coupling, thus rationalizing a qualitative trend between $C_{2+}$ production and GB density.

## Results

### CO electrolysis performance

The porous nanocrystalline Cu nanoparticle (Cu-nc) catalyst with high-density GBs was synthesized by reducing $CuCl_2$ with $NaBH_4$ in the absence of any additives at room temperature. The CO electrolysis performance of the Cu-nc catalyst was measured in a home-made zero-gap alkaline MEA electrolyzer with an electrode area of $4\,cm^2$ (Supplementary Fig. 2) described previously[15]. The CO electrolysis was performed in the galvanostatic mode. The anode and cathode were fed with 0.5 M KOH solution at a flow rate of $5\,mL\,min^{-1}$ and dry CO at a flow rate of $80\,mL\,min^{-1}$, respectively. The Cu-nc powder catalyst was incorporated into a gas diffusion electrode (GDE) with polytetrafluoroethylene (PTFE) as a binder in the catalyst layer. The hydrophobic and porous GDE structure drastically reduces the diffusion pathway for CO to reach the catalyst, resulting in high current densities[15,16]. Moreover, through careful optimization in the assembly and operation[17], the MEA electrolzyer used in this work exhibits an Ohmic resistance as low as $0.13\,\Omega\!\cdot\!cm^2$ (Supplementary Fig. 3), which is very important for reducing cell voltage and increasing full-cell energy efficiency. The high performance of our MEA electrolyzer has been demonstrated using commercially available Cu nanoparticles (Supplementary Fig. 4). As shown in Fig. 1a, CO is selectively reduced to $C_{2+}$ products including ethylene, ethanol, acetate, and n-propanol, while no $C_1$ products like $CO_2$ and methane are detected. The $C_{2+}$ FE is up to over 90%, while the $H_2$ FE is as low as $2.03 \pm 0.68\%$ (Fig. 1a). More remarkably, while the $C_{2+}$ FE slightly decreases to $87 \pm 1\%$, a high total current density of $5.0\,A\,cm^{-2}$ is achieved at a low cell voltage of $2.78 \pm 0.01\,V$ (Supplementary Fig. 5), resulting in a notable $C_{2+}$ partial current density of $4.35 \pm 0.07\,A\,cm^{-2}$. The corresponding $C_{2+}$ and ethylene formation rates reach $0.39 \pm 0.01\,mmol\,min^{-1}\,cm^{-2}$ and $3.44 \pm 0.12\,mL\,min^{-1}\,cm^{-2}$ (Fig. 1b). The CO electrolysis performance in terms of $C_{2+}$ FE and partial current density is well-placed among previous reports (Fig. 1c, Supplementary Table 2)[8,9,15,18–24]. As no $C_1$ by-products are generated, the $C_{2+}$ carbon selectivity is -100%, even at a high CO conversion of $85 \pm 3\%$ and a high converted CO rate of $65.1 \pm 2.3\,mL\,min^{-1}$, is favorable compared to reported thermocatalytic CO hydrogenation processes (Fig. 1d,e, Supplementary Table 3)[3–5,25–32]. Furthermore, the full-cell energy efficiency towards CO electrolysis to $C_{2+}$ products is above 32%, with a peak value of $39.6 \pm 0.5\%$ at a total current density of $3.0\,A\,cm^{-2}$ (Supplementary Fig. 6). The stability of the Cu-nc catalyst was measured at a high applied current density of $1.0\,A\,cm^{-2}$. Over a course of 150 h, the cell voltage only increases by 0.12 V. The ethylene FE is almost stable, and the $C_{2+}$ FE slightly decreases but is still above 83.6% (Fig. 1f). The $H_2$ FE gradually increases to 16.7%, which is likely attributed to slow electrode flooding due to the loss of hydrophobicity over time as indicated by contact angle measurements before and after the stability test (Supplementary Fig. 7). Nevertheless, we demonstrate the great promise for highly efficient electrochemical synthesis of $C_{2+}$ chemicals from CO using the GB-rich Cu-nc catalyst.

### Apparent trend between $C_{2+}$ production and GB density

To reveal structure-reactivity relations of the Cu-nc catalyst for the notable CO electrolysis performance, thorough characterizations and control experiments were conducted. The Cu-nc catalyst is highly

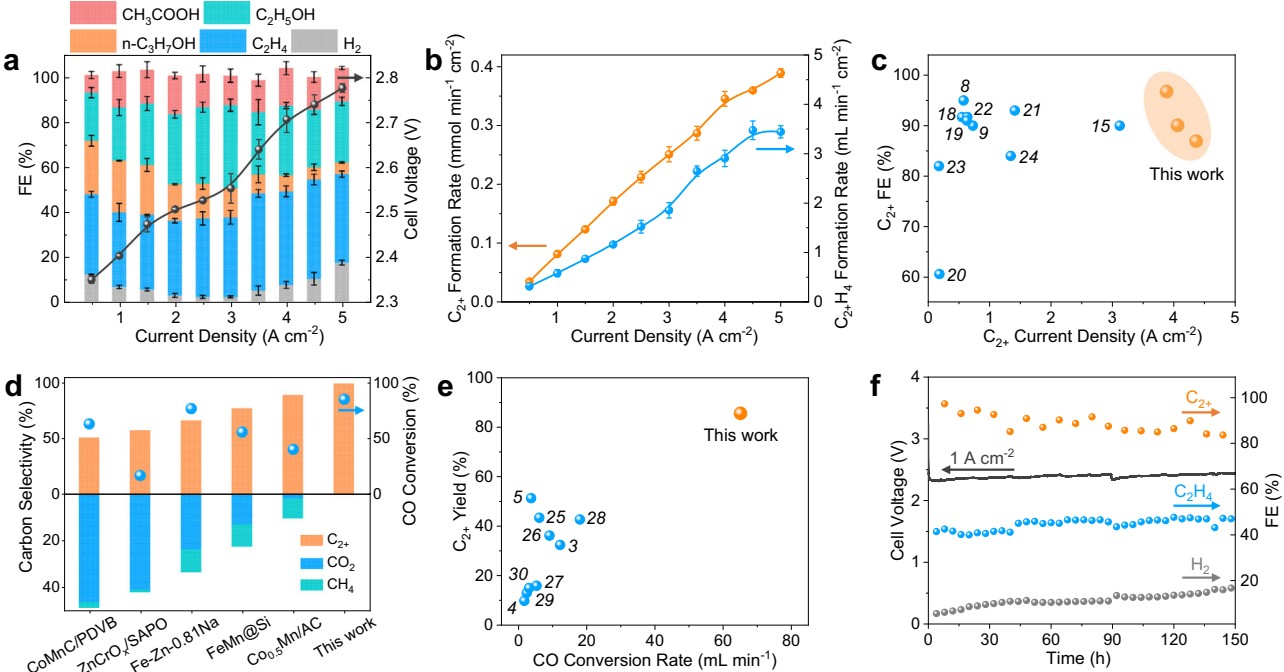

**Fig. 1 | CO electrolysis performance over Cu-nc catalyst. a** Faradaic efficiencies (FEs) and cell voltage and (b) ethylene and $C_{2+}$ formation rates as a function of current density. The error bars represent standard error of the mean and are made based on three fully separate and identical measurements. **c** CO electrolysis performance comparison[8,9,15,18–24]. **d, e** Performance comparison between CO electrolysis in this work and thermocatalytic CO hydrogenation[3–5,25–30]. **f** Stability test at a current density of $1.0\,A\,cm^{-2}$. Source data are provided as a Source Data file.

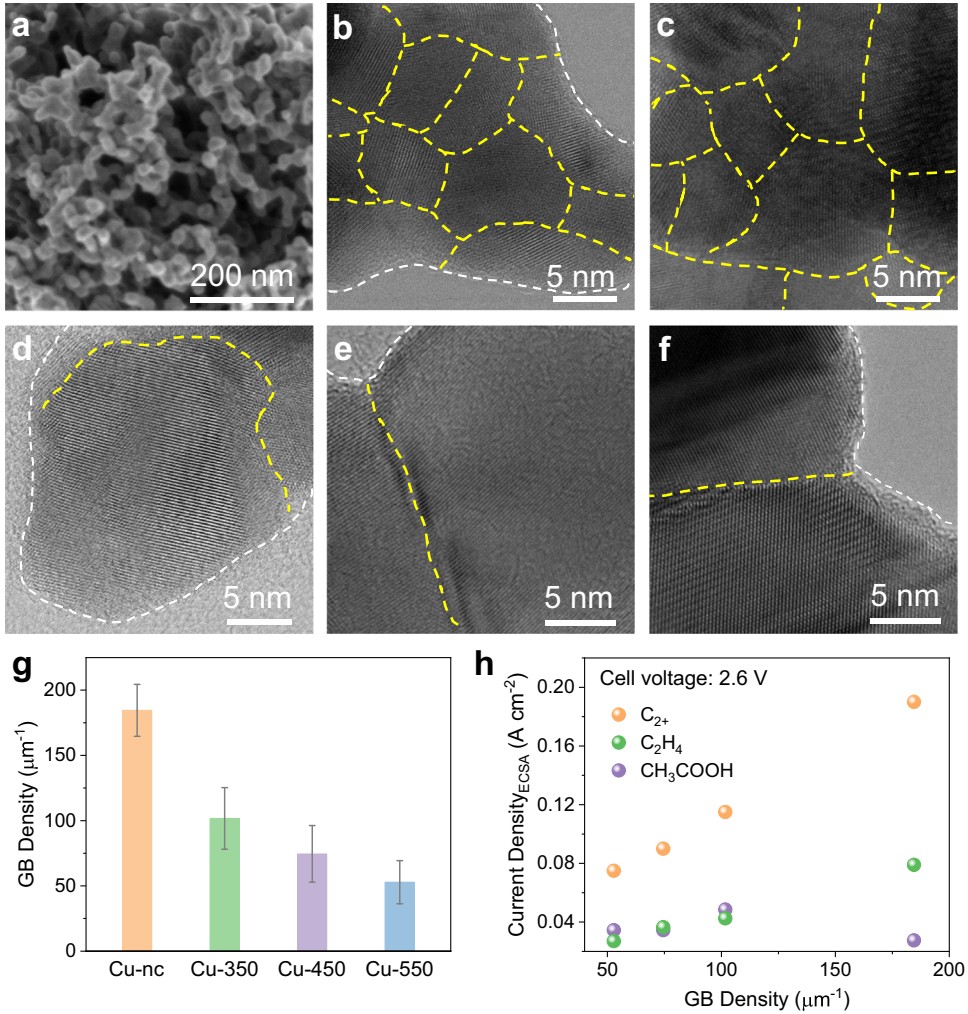

**Fig. 2 | Correlation between $C_{2+}$ production and grain boundary (GB) density.** **a** SEM image of Cu-nc catalyst. HRTEM images of (**b**) Cu-nc, (**c**) Cu-nc after CO electrolysis, (**d**) Cu-350, (**e**) Cu-450, and (**f**) Cu-550 catalysts. **g** GB densities of Cu-nc, Cu-350, Cu-450, and Cu-550 catalysts after electrolysis. The error bars represent standard error of the mean and are made based on three fully separate and identical measurements. **h** Correlations between electrochemically active surface area (ECSA)-normalized $C_{2+}$/ethylene/acetate partial current densities and GB density after electrolysis at a cell voltage of 2.6 V. Source data are provided as a Source Data file.

porous with interconnected nanocrystalline networks as shown in both scanning electron microscopy (SEM) and transmission electron microscopy (TEM) images (Fig. 2a, Supplementary Figs. 8,9). High-resolution TEM (HRTEM) images (Fig. 2b, Supplementary Figs. 10,11) show the presence of high-density GBs in the Cu-nc catalyst. The as-prepared Cu-nc catalyst was further annealed in air at 350, 450 and 550 °C for 2 h to reduce GB density[33], and the treated samples were denoted as Cu-$x$ ($x$ is 350, 450, and 550, respectively). While the porous structure remains over the Cu-$x$ catalysts after annealing in air, the number of GBs drastically decreases (Fig. 2d–f, Supplementary Figs. 10,12–14). The densities of GBs present in the Cu-nc and Cu-$x$ catalysts were quantified by analyzing ten typical HRTEM images for each sample (Supplementary Figs. 11–18). The Cu-nc catalyst has a GB density of $204.2 \pm 25.3 \, \mu m^{-1}$, 5-fold higher than previously reported carbon-supported Cu nanoparticles[33]. Such a high GB density is ascribed to the interconnected networks comprised of nanosized Cu domains (Fig. 2a, Supplementary Figs. 8, 9). From the statistical results (Fig. 2g and Supplementary Tables 4−6), the GB density decreases with increasing annealing temperature, and it only changes slightly after CO electrolysis. Meanwhile, the annealing treatment transforms the partially oxidized Cu-nc catalyst with mixed Cu and $Cu_2O$ phases into fully oxidized Cu-$x$ catalysts with pure CuO phase, as demonstrated by X-ray diffraction (XRD) and X-ray photoelectron spectroscopy (XPS) results

(Supplementary Figs. 19, 20). OH⁻ adsorption spectra measurements conducted in 1 M KOH show that the three $OH_{ad}$ peaks at 0.44, 0.39, and 0.34 V vs. reversible hydrogen electrode, assigned to the (111), (110) and (100) Cu facets, respectively, have very similar intensities for the Cu-nc and Cu-$x$ catalysts (Supplementary Fig. 21). No any $OH_{ad}$ peaks appear at a more negative potential, ruling out the existence of high-index facets on these catalysts[21,34,35]. The electrochemically active surface areas (ECSAs) of these catalysts determined by Pb under-potential deposition (UPD) measurements are close to each other but slightly decrease after annealing treatment (Supplementary Fig. 22, Supplementary Table 7). The CO electrolysis performances of the Cu-$x$ catalysts are shown in Supplementary Figs. 23, 24. Compared to the Cu-nc catalyst, the geometric current densities of the Cu-$x$ catalysts are lower, while the product selectivity shifts from ethylene towards acetate (Supplementary Fig. 25). We plot the ECSA-normalized partial current densities of $C_{2+}$, ethylene, and acetate at a fixed cell voltage (2.5, 2.6, and 2.7 V) as a function of GB density. It is clear that the ECSA-normalized $C_{2+}$ partial current density shows very positive correlations with GB density after electrolysis (Fig. 2h, Supplementary Fig. 26). More specifically, the production of ethylene, ethanol and n-propanol increases with increasing GB density, while the acetate production seems to be independent on GB density (Fig. 2g, Supplementary Figs. 26, 27). As the initial Cu oxidation states in the Cu-nc and Cu-$x$

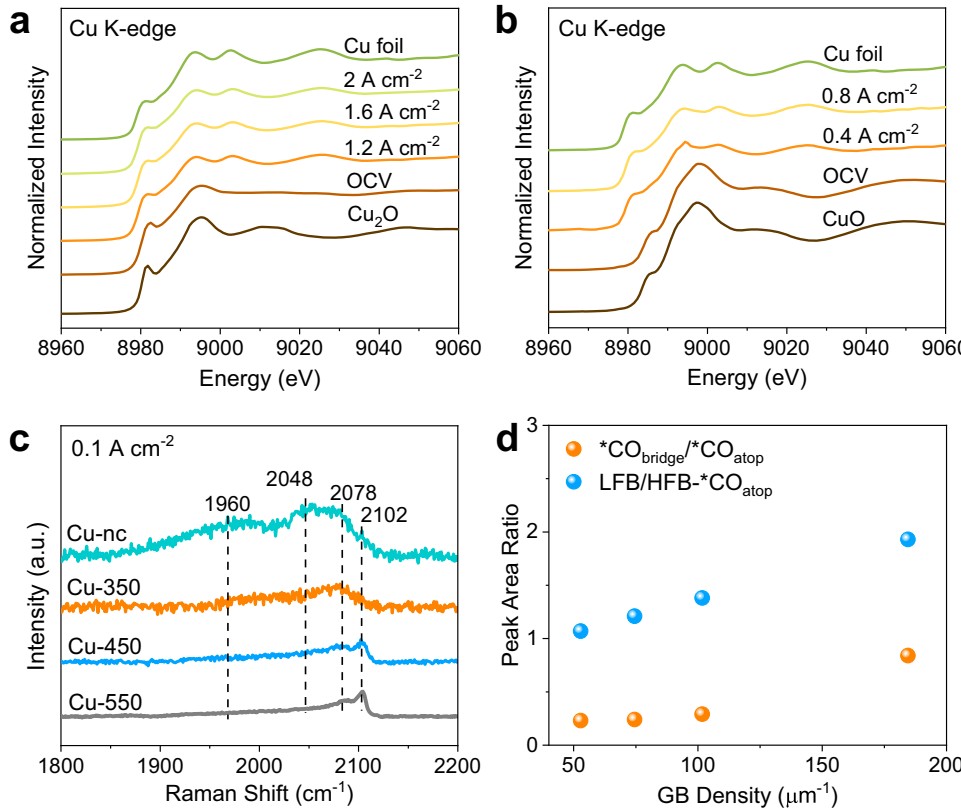

**Fig. 3 | Operando spectroscopy studies.** Operando Cu K-edge XANES measurements over Cu-nc (**a**) and Cu-350 (**b**) catalysts. **c** Operando Raman spectra for *CO$_{bridge}$ and *CO$_{atop}$ over Cu-nc, Cu-350, Cu-450, and Cu-550 catalysts at 0.1 A cm$^{-2}$. **d** Ratios of *CO$_{bridge}$/*CO$_{atop}$, LFB/HFB *CO$_{atop}$ versus grain boundary (GB) density after electrolysis at 0.1 A cm$^{-2}$. Source data are provided as a Source Data file.

catalysts are different (XRD and XPS results in Supplementary Figs. 19, 20), the differences in the production of C$_{2+}$ products over these catalysts are also likely caused by Cu oxidation state, in addition to GB density.

## Cu oxidation state

Operando spectroscopic characterizations were further conducted to track the oxidation state of Cu catalysts under reaction conditions[35,36]. A home-made MEA operando cell used in this work (Supplementary Figs. 28, 29) allows us to perform spectroscopic characterizations under very similar reaction conditions where the performance data at a current density of hundreds of mA cm$^{-2}$ are acquired. Figure 3a, b show Cu K-edge X-ray absorption near edge structure (XANES) spectra of the Cu-nc and Cu-350 catalysts at the open circuit voltage (OCV) and different applied current densities. The Cu-nc catalyst shows mixed Cu and Cu$_2$O phases at OCV, in consistent with XRD results. On applying current densities for several minutes, the Cu-nc catalyst is electrochemically reduced to metallic Cu (Fig. 3a). While the Cu-350 catalyst displays a CuO phase at OCV, metallic Cu is present during CO electrolysis as well (Fig. 3b). Quasi in situ XPS measurements without air exposure indicate that the surfaces of both Cu-nc and Cu-350 catalysts after CO electrolysis show the presence of similar amounts of Cu$^+$ species (Supplementary Fig. 20). Overall, the oxidation states of the Cu-nc and Cu-350 catalysts are almost same under CO electrolysis conditions. Therefore, the role of initial Cu oxidation state in C$_{2+}$ production is excluded and the GBs are very likely the active sites for CO electrolysis to C$_{2+}$ products.

## Operando Raman spectroscopy studies

To provide in-depth insights into the role of GBs in C–C coupling, surface adsorbed intermediates during CO electrolysis were studied via operando Raman spectroscopy (Supplementary Fig. 29)[37]. Generally, the peaks of atop-adsorbed and bridge-adsorbed CO (*CO$_{atop}$ and *CO$_{bridge}$) are observed at 1900–2100 cm$^{-1}$, when current densities are applied to the Cu-nc and Cu-x catalysts (Supplementary Fig. 30). The broad *CO peak can be deconvoluted to high-frequency-band (HFB)-*CO$_{atop}$ at 2102 and 2078 cm$^{-1}$, low-frequency-band (LFB)-*CO$_{atop}$ at 2048 cm$^{-1}$, and *CO$_{bridge}$ at 1960 cm$^{-1}$ (Supplementary Fig. 31)[38,39]. The presence of *CO$_{bridge}$ was further confirmed by attenuated total reflectance Fourier transform infrared (ATR-FTIR) spectroscopy measurements (Supplementary Fig. 32). Figure 3c shows the Raman spectra over the Cu-nc and Cu-x catalysts at 0.1 A cm$^{-2}$. The *CO peaks shift to higher vibration frequencies with increasing annealing temperature (thus, decreasing GB density, Supplementary Table 8). Figure 3d plots the ratios of *CO$_{bridge}$/*CO$_{atop}$ and LFB/HFB-*CO$_{atop}$ versus GB density. Both ratios increase with increasing GB density, indicating that *CO binding over the Cu-nc catalyst is stronger than that over the Cu-x catalysts[40,41]. These *CO peaks get weaker with increasing current densities, due to the lowered *CO coverage caused by accelerated *CO conversion. However, at a higher current density, e.g., 0.3 A cm$^{-2}$, *CO is hardly observed over the Cu-x catalysts, but still visible over the Cu-nc catalyst (Supplementary Fig. 30). Therefore, the *CO coverage during CO electrolysis is also higher over the Cu-nc catalyst versus the Cu-x catalysts. Overall, it is reasonable to postulate that GBs facilitate *CO binding and improve its coverage, thus enhancing subsequent C–C coupling.

## DFT calculations

The role of the GBs in promoting C–C coupling and tuning the selectivity among C$_{2+}$ products was further investigated using DFT calculations. The GBs were simulated following the coincidence site lattice (CSL) theory[42]. As the ratios of the Cu(111), Cu(110), and Cu(100)

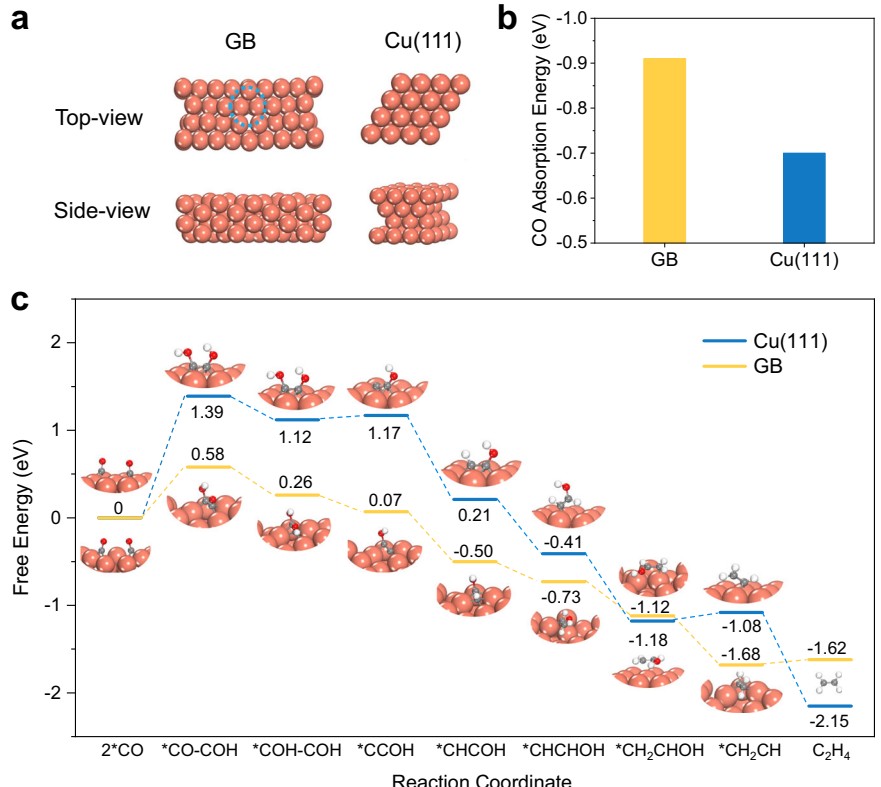

**Fig. 4 | Density functional theory (DFT) calculations. a** Atomic configuration, (**b**) CO adsorption energy, and (**c**) free energy profile for ethylene production on grain boundary (GB) and Cu(111). Source data are provided as a Source Data file.

facets were similar in the Cu-nc and Cu-x catalysts (Supplementary Fig. 21), we built three CSL GBs for Cu(111), Cu(110), and Cu(100) facets, respectively (Fig. 4a and Supplementary Figs. 33a, 34a). Compared to flat Cu(111) surface, CO adsorption on GBs is significantly improved, with a CO adsorption energy of −0.70 eV on Cu(111) and −0.91 eV on GBs (Fig. 4b, Supplementary Table 9). The improved CO adsorption on GBs is beneficial to increase the surface coverage of *CO for subsequent reactions[43,44]. The energy profiles towards ethylene formation on GBs and Cu(111) are shown in Fig. 4c. The *COCOH formation via C − C coupling is the most demanding energetically. The free energy changes of this step are 0.58 and 1.39 eV on GBs and Cu(111), which indicates that GBs are much more active for C − C coupling. Furthermore, the energy profile indicates that other steps along the pathway exhibit a generally downhill trend on GBs, which is beneficial for the production of ethylene. We further investigated the *CO adsorption and energy profiles on Cu(110), Cu(100), and their corresponding GBs, which indicates that C − C coupling reaction is significantly improved on GBs (Supplementary Figs. 33, 34). In contrast, the acetate production is more favorable on Cu(111) over GBs (Supplementary Fig. 35). Thus, the ethylene pathway is preferred on GBs compared to Cu(111). Overall, the stronger *CO adsorption and lower C − C coupling reaction energy on GBs improve $C_{2+}$ production and the selectivity of ethylene versus acetate. These calculation results explain well the experimentally observed positive qualitative trends between CO electrolysis performance and GB density (Fig. 2h).

## Scale-up demonstration of CO electrolysis

To validate the feasibility for large-scale electrochemical synthesis of $C_{2+}$ chemicals from CO using the Cu-nc catalyst, we first scaled up the CO electrolysis process using a 100 cm² MEA electrolyzer (Fig. 5a and Supplementary Fig. 36). The anode and cathode were fed with 0.5 M KOH solution at a flow rate of 0.125 L min⁻¹ and dry CO at a flow rate of 2.0 L min⁻¹, respectively. Figure 5b show the CO electrolysis

performance at an applied total current of 100, 200, 300, 400, and 500 A. The $C_{2+}$ FE is above 92% at 100 − 300 A and decreases to 86% at 400 A and 73% at 500 A. The stability test conducted at an applied total current of 100 A (1.0 A cm⁻²) shows that the cell voltage is stable at around 2.5 V and the $C_{2+}$ FE maintains above 88% over a course of 32 h (Fig. 5c). An electrolyzer stack with five 100-cm² MEAs was further assembled (Fig. 5d, Supplementary Figs. 37, 38). The anode and cathode were fed with 0.5 M KOH solution at a flow rate of 0.65 L min⁻¹ and dry CO at a flow rate of 10.0 L min⁻¹, respectively. Figure 5e, f show the stack performance of the Cu-nc catalyst at a total current of 100, 200, 300, and 400 A. The $C_{2+}$ FE is above 96% at 100 − 200 A and decreases to 84% at 300 A and 64% at 400 A (Fig. 5e). The highest $C_{2+}$ and ethylene formation rates reach 118.9 mmol min⁻¹ and 1.2 L min⁻¹ (Fig. 5f). Remarkably, the maximum power (electrolysis scale) of the stack reaches as high as 5.8 kW at 400 A. While further efforts should be input in the future to improve the effectiveness and long-term stability of the scale-up process, these scale-up attempts indicate that CO electrolysis is a very promising and practical route for the electrochemical synthesis of valuable $C_{2+}$ chemicals. The CO electrolysis process can be economically viable as demonstrated by techno-economic assessment (TEA) based on the represented performance data at 1.0, 3.0, and 4.5 A cm⁻² using electricity derived from renewable energy (Supplementary Note 1 and Supplementary Fig. 39), and the $CO_2$ emission from CO electrolysis can be reduced by up to 82% compared to thermocatalytic CO hydrogenation via Fischer-Tropsch synthesis (Supplementary Note 2 and Supplementary Fig. 39).

## Discussion

In summary, we demonstrate an electrochemical route for highly efficient synthesis of $C_{2+}$ chemicals from CO with the GB-rich Cu nanoparticle catalyst. We present a notable CO electrolysis performance with a $C_{2+}$ partial current density as high as 4.35 ± 0.07 A cm⁻² at a low cell voltage of 2.78 ± 0.01 V in a home-made alkaline MEA

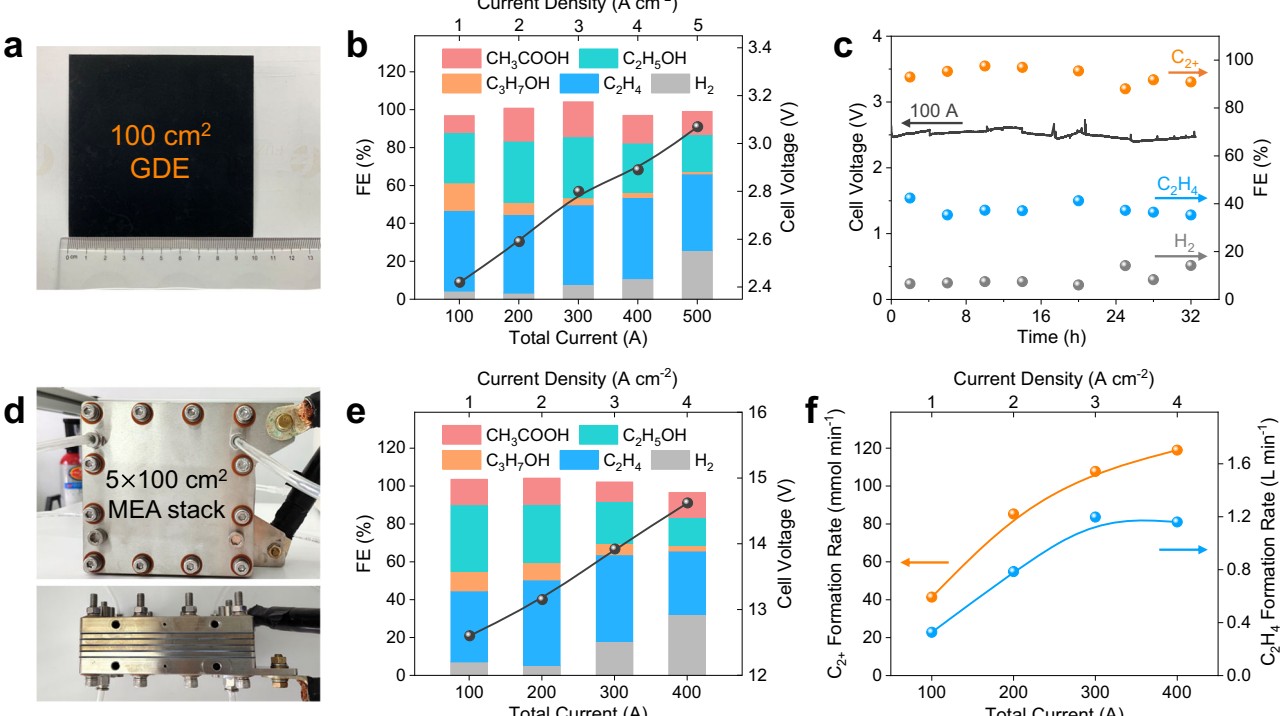

**Fig. 5 | Scale-up demonstration of CO electrolysis. a** Photograph of 100 cm² gas diffusion electrode (GDE). **b** Faradaic efficiency (FE) and cell voltage as a function of total current over Cu-nc catalyst in the 100 cm² membrane electrode assembly (MEA) electrolyzer. **c** Stability test at a total current of 100 A in the 100 cm² MEA electrolyzer. **d** Photographs of electrolyzer stack with five 100 cm² MEAs. **e** FE and cell voltage as well as (**f**) C₂₊ and C₂H₄ formation rates as a function of total current. Source data are provided as a Source Data file.

electrolyzer. CO is exclusively converted to C$_{2+}$ products (~100% carbon selectivity) and its conversion reaches up to 85 ± 3% at a high converted CO rate of 65.1 ± 2.3 mL min$^{-1}$. Operando spectroscopy characterization and DFT calculation studies reveal the role of the GBs of Cu nanoparticles in the improved C$_{2+}$ production. A scale-up demonstration using an electrolyzer stack with five 100 cm² MEAs at an applied current of 400 A achieves high formation rates of C$_{2+}$ products and ethylene with 118.9 mmol min$^{-1}$ and 1.2 L min$^{-1}$, respectively, highlighting the great promise of CO electrolysis as a practical route for the electrochemical synthesis of C$_{2+}$ valuable chemicals.

## Methods
### Chemicals and materials
Copper(II) chloride (CuCl₂), sodium borohydride (NaBH₄), polytetra-fluoroethylene (PTFE, 60 wt% dispersion in H₂O) suspension were purchased from Sigma-Aldrich. Potassium hydroxide (KOH) was purchased from Aladdin. Pb(ClO₄)₂·3H₂O was purchased from Macklin. Cu foil (99.9%, 0.127 mm thick) was purchased from Alfa Aesar. Ir black catalyst was purchased from Johnson Matthey Corp. Cu nanoparticles (product no. 774081) were purchased from Sigma-Aldrich. Ultrapure water (18.2 MΩ) was used in all experiments. All the chemicals were used without further purification.

### Catalyst synthesis
Cu-nc catalyst was synthesized in the following procedure. 10 mmol CuCl₂ powder was dispersed in 300 mL water at 700 rpm. Then 50 mM NaBH₄ (dissolved in 50 mL water) was added dropwise in 2 min and the mixture was continuously stirred for 20 min. Then, the black precipitates were collected by filtration and washed with de-ionized water and ethanol, and finally dried in vacuum. Cu-350, Cu-450, and Cu-550 catalysts were prepared by annealing the as-prepared Cu-nc catalyst in air at 350, 450, and 550 °C for 2 h.

### Preparation of gas diffusion layer (GDL)
Firstly, Vulcan XC-72R carbon black was dispersed in ethanol, and certain amount of PTFE suspension was added with mechanically stirring to form a homogeneous carbon black ink. Then, the ink was hand-painted onto one side of carbon paper (Toray TPG-H-60) and was annealed in air at 350 °C for 1 h in a muffle furnace to obtain the final GDL. The carbon black loading was about 1.0 mg cm$^{-2}$ and the PTFE content in the GDL was 15 wt%.

### Preparation of GDE
The Cu-nc or Cu-$x$ catalysts and PTFE solution were dispersed in ethanol with a mass ratio of 3:1 to form an ink. The ink was then painted onto the GDL to form a GDE. The catalyst mass loading was 2.0 ± 0.1 mg cm$^{-2}$.

### Preparation of anode
Commercial Ir black catalyst was dispersed in ethanol, and certain amount of quaternary ammonia poly(*N*-methyl-piperidine-*co*-*p*-ter-phenyl) (QAPPT) ionomer solution was added with mechanically stirring to form a homogeneous ink. Then, the ink was drop-casted onto a Ti foam to form an anode. The Ir black loading was 1.0 ± 0.1 mg cm$^{-2}$ and the QAPPT content in the anode catalyst layer was 10 wt%.

### Material characterization
The powder XRD patterns were recorded with a PANalytical X'pert PPR diffractometer with a Cu *K*α radiation source (λ = 1.5418 Å) at 40 kV and 40 mA at a scan rate of 8° min$^{-1}$. The morphologies of the catalysts were acquired using a field emission scanning electron microscopy (FE-SEM, JSM-7800F) with an accelerating voltage of 3 kV. TEM and HRTEM images were acquired by a JEM-2100 microscopy and a JEM-ARM300F microscopy with an accelerating voltage of 200 and 300 kV, respectively. XPS spectra were recorded on a Thermo Scientific ESCALAB 250Xi spectrometer with an Al *K*α X-ray source. All the

binding energies were calibrated with C 1s spectrum with peak intensity at 284.8 eV.

## CO electrolysis measurements

CO electrolysis experiments were performed at ambient temperature ($20 - 25\ ^{\circ}$C) in a home-made alkaline MEA electrolyzer with an electrode area of 4 cm$^2$ as described previously[15]. Anion exchange membranes (QAPPT) were synthesized with reference to a previous article[45]. The electrolyzer was assembled using two Pt-coated titanium flow field plates for CO feeding at the cathode, aqueous solution feeding at the anode, as well as for current collecting, respectively. A fresh catalyst-coated cathode, a QAPPT membrane and an Ir-coated anode were used for each electrolysis test. The cathodic flow field was fed with dry 95% CO/5% N$_2$ (here N$_2$ as an internal standard for quantification, and for simplicity CO was used to represent 95% CO/5% N$_2$ in the main text) at a flow rate of 80 mL min$^{-1}$ through a mass flow controller. The anodic flow field was fed with 0.5 M KOH solution at a flow rate of 5.0 mL min$^{-1}$ using a peristaltic pump. The electrolysis was carried out in the galvanostatic mode using an Autolab potentiostat/galvanostat (PGSTAT 302 N with 10.0 A booster), and an Autoranging System DC Power Supply (Keysight N8940A, 0-80 V/0-170 A, 5000 W) while the current was greater than 10.0 A. The scale-up measurements using a 100 cm$^2$ MEA electrolyzer and an electrolyzer stack with five 100 cm$^2$ MEAs were carried out in the galvanostatic mode using an Autoranging System DC Power Supply (Keysight N8951A, 0-80 V/0-510 A, 15000 W), and the cathode was fed with dry 95% CO/5% N$_2$ at a flow rate of 2.0 and 10.0 L min$^{-1}$, respectively. For the 100 cm$^2$ electrolyzer and the electrolyzer stack, the anodes were fed with 0.5 M KOH solutions at a flow rate of 0.125 and 0.65 L min$^{-1}$, respectively. In all above CO electrolysis experiments including stability measurements, the anolyte was not recirculated, and fresh anolyte was always used.

## Product analysis

Gas products were analyzed by an on-line gas chromatography (Shimadzu, GC-2014) equipped with a thermal conductivity detector (TCD) and a flame ionization detector (FID). Liquid products collected from anolyte were analyzed by a Bruker AVANCE III 400 MHz nuclear magnetic resonance (NMR) spectrometer. A mixture of anolyte and 1-propanesulfonic acid 3-(trimethylsilyl) sodium salt (DSS, as an internal standard for quantification) was used for NMR measurements. The one dimensional $^1$H-NMR spectrum was measured with water suppression using a pre-saturation method.

The Faradaic efficiency of a specific product is calculated as follows:

$$\varepsilon_{Faradaic,i} = Q_i/Q_{total} \times 100 = (N_i \times n_i \times F)/Q_{total} \times 100 \quad (1)$$

Where,

$\varepsilon_{Faradaic,i}$: the Faradaic efficiency of product i, %;
$Q_{total}$: the consumed charge, C;
$Q_i$: the charge used for the formation of the product i, C;
$N_i$: the amount of the product i, mol;
$n_i$: the number of electrons transferred to form the product i;
F: Faraday constant, which is 96485 C mol$^{-1}$.

Partial current density of a specific product is calculated as follows:

$$j_{partial,i} = j_{total} \times \varepsilon_{Faradaic,i} \quad (2)$$

Normalized current density is calculated as follows:

$$j_{norm} = j_{geometric}/RF (RF \text{ is the roughness factor of a given electrode}) \quad (3)$$

The energy efficiency for the formation of a specific product is defined as follows:

$$\varepsilon_{Energy,i} = \frac{\Delta H_i^0}{\Delta G_i} \times \varepsilon_{Faradaic,i} = \frac{n_i \times F \times E^n}{n_i \times F \times E_i} \times \varepsilon_{Faradaic,i} = \frac{E^n}{E_i} \times \varepsilon_{Faradaic,i} \quad (4)$$

Where,

$\varepsilon_{Energy,i}$: the energy efficiency for the formation of product i, %;
$\Delta H_i^0$: the theoretical enthalpy change of product i, kJ mol$^{-1}$;
$\Delta G_i$: the changes in the Gibbs free energy of product i, kJ mol$^{-1}$;
$\varepsilon_{Faradaic,i}$: the Faradaic efficiency of product i, %;
$n_i$: the number of electrons transferred to form the product i;
F: Faraday constant, which is 96485 C mol$^{-1}$;
$E^n$: the thermoneutral voltage (calculated from $\Delta H_i^0$), V;
$E_i$: the applied cell voltage, V.

The energy efficiency of total CO electrolysis products reported in this work is the sum of that of each individual product.

The error bars in reporting Faradaic efficiency, energy efficiency, and cell voltage in this work represent the standard deviation from three fully separate and identical measurements.

## ECSA measurements

Pb UPD was performed to determine the ECSAs of the catalysts deposited on the GDE. After the catalysts were electrochemically reduced, cyclic voltammetry (CV) measurements were carried out at a scan rate of 10 mV s$^{-1}$ in a solution containing 0.1 M HClO$_4$ and 0.001 M Pb(ClO$_4$)$_2$. Prior to the CV measurements, the solution was purged with Ar for at least 30 min. Here, Cu foil was used for reference according to previous literature[46]. The measurements were conducted at ambient temperature ($20 - 25\ ^{\circ}$C) and no iR correction was performed.

## OH$^-$ adsorption measurements

CVs for OH$^-$ adsorption measurements were recorded in H-cell using Ar-purged 1 M KOH as electrolyte and Ag/AgCl as reference electrode after the catalysts were electrochemically reduced. A potential window from $-0.2$ to 0.55 V (vs. RHE) and a scan rate of 20 mV s$^{-1}$ were selected during OH$^-$ adsorption measurements. The measurements were conducted at ambient temperature ($20 - 25\ ^{\circ}$C) and no iR correction was performed.

## Quantification of GB density from TEM images

GB densities of Cu-nc, Cu-350, Cu-450, and Cu-550 catalysts were measured using the method described below. GBs are considered as the border of two regions with different lattice orientations and are marked with yellow dashed lines in the TEM images. For each sample, ten typical TEM images are analyzed. GB density is defined as the GB length per unit area of Cu nanoparticle surface. The length of GBs ($L$) and nanoparticle area ($S_i$) in each TEM image was quantified by Gatan DigitalMicrograph. The length of GBs ($L$) was defined as the total length of the yellow dashed lines. The nanoparticle area ($S_i$) was calculated by the difference between the total area and the blank area of an image. Assuming that the surface GB density is calculated using the following equation:

$$\frac{Grain\ boundary\ length}{Nanoparticle\ surface\ area} = \frac{\sum L}{\sum S_i} \quad (5)$$

## Contact angle measurements

Contact angle measurements were conducted by a DSA100 Drop Shape Analyzer. Video was recorded when water was being pumped to the drop slowly from the syringe via the needle, and the water front advances on the sample. Each image of this video was later analyzed to determine the contact angle when the image was captured.

## Operando X-ray adsorption spectroscopy (XAS) measurements

The measurements at Cu K-edge ($E_0$ = 8979 eV) were carried out in fluorescence mode using a Lytle detector at the BL11B beamline of the Shanghai Synchrotron Radiation Facility. The energy was calibrated to the absorption edge of a Cu foil. The CO electrolysis was performed in 0.5 M KOH with a reactant gas flow rate of 5.0 mL min$^{-1}$ in a modified MEA cell in the galvanostatic mode. The measurements were conducted at ambient temperature (20 − 25 °C) and no iR correction was performed. The gas chamber had a small window cut out and sealed with Kapton film to allow fluorescence signals to pass from the electrode to the detector. The XAS data were processed using the software package Athena and ARTEMIS.

## Quasi in situ XPS measurements

CO electrolysis experiments were firstly performed in the MEA electrolyzer in glovebox. During CO electrolysis, the produced $O_2$ were expelled out of glovebox, and the concentration of $O_2$ in glovebox kept below 0.01 ppm. After CO electrolysis at 2.6 V for 1 h, the electrodes were transferred by a mobile transfer chamber to the XPS analysis chamber. The measurements were conducted at ambient temperature (20 − 25 °C) and no iR correction was performed. The sample was kept in inert atmosphere or vacuum during the entire transfer process without exposure to air. The XPS spectra were recorded on a Thermo Scientific ESCALAB 250Xi spectrometer with an Al $K\alpha$ X-ray source operated at 300 W. All the binding energies were calibrated with C 1 s spectrum with peak intensity at 284.8 eV.

## Operando Raman spectroscopy measurements

Operando Raman spectroscopy measurements were carried out using a Renishaw inVia Raman microscope in a homemade MEA cell which was similar to the modified MEA cell for in situ XAS measurements. The measurements were conducted at ambient temperature (20 − 25 °C) and no iR correction was performed. A near-infrared laser (785 nm) was used as the excitation source. A long focal length objective lens (Leica, 50×) was used for focusing and collecting the incident and scattered laser light. A fresh catalyst-coated cathode, a QAPPT membrane and an Ir-coated anode were used for each test. The Cu-nc and Cu-x catalysts were painted onto one side of the QAPPT membrane which served as the cathode. To get steady-state Raman spectra, Raman signals were collected after reduction for 10 min at each applied current density.

## In situ ATR-IR measurements

The catalyst ink was drop-casted via pipette onto a hemicylindrical silicon prism covered with three layers of graphene. A Pt foil and a saturated calomel electrode (SCE) electrode were used as counter and reference electrodes, respectively. The electrolyte was 0.1 M KOH, which was constantly purged with CO during the experiment. Before the experiments, the working electrode was reduced to stable state by continuously scanning. The electrode potential was held at 0.3 V vs. RHE, and a background spectrum (reflectance $R_0$) was recorded. The electrode potential was altered stepwise from −0.1 to −0.6 V vs. RHE, and in the meantime IR spectra were recorded with a time resolution of 42 s per spectrum at a spectral resolution of 8 cm$^{-1}$. The measurements were conducted at ambient temperature (20 − 25 °C) and no iR correction was performed. All spectra were reported as the relative change in reflectivity, $\Delta R/R_0 = (R − R_0)/R_0$, where R and $R_0$ are single-beam spectra collected at the sample potential and the reference potential, respectively. A Nicolet 8700 infrared spectrometer with a HgCdTe detector cooled by liquid nitrogen was used.

## Theoretical calculations

DFT calculations were performed using the Vienna ab initio simulation package (VASP)[47,48]. The revised Perdew-Burke-Ernzerhof functional (RPBE) from Hammer et al. was employed for electron exchange−correlation[49,50]. The electron-ion interactions were described by projector augmented wave potentials proposed by Blochl and implemented by Kresse[51,52]. The plane wave basis set with an energy cutoff of 400 eV was used for geometry optimizations. Spin-polarized calculations were conducted using gamma-centralized grid of k-points of 4 × 4 × 1 for Cu(111), 2 × 2 × 1 for model of Cu grain boundary, respectively. For all the calculations, the van der Waals (vdW) contributions were evaluated with a DFT-D3 method[53]. The electronic energy and forces were converged to within 1 × 10$^{-6}$ eV and 0.02 eV/Å. The vertical vacuum slab was set to be at least 10 Å in all cases. We built three representative Cu grain boundaries, Cu$\Sigma$3/(111), Cu$\Sigma$5/(100), and Cu$\Sigma$3/(110) with Aimsgb code by Yang et al. [42]. The low $\Sigma$ values were chosen for construction, because low $\Sigma$ boundaries tended to have lower energies than average[54]. We acknowledge the existence of various grain boundary types, and modeling all types of grain boundaries is not feasible[55]. We expect that the use of three representative models, Cu$\Sigma$3/(111), Cu$\Sigma$5/(100), and Cu$\Sigma$3/(110), which were referred to as Cu(111)-GB, Cu(100)-GB and Cu(110)-GB, could effectively represent the essential characteristics of grain boundaries, and shed light on how grain boundaries influence catalytic performance compared to flat (111), (100), and (110) facets. The Cu(111)-GB, Cu(100)-GB and Cu(110)-GB are consisted with 48, 64, 60 copper atoms as shown in Fig. 4a and Supplementary Figs. 33a, 34a. The bottom two layers of the copper models were fixed and the other atoms were permitted to relax. In order to correct the significant self-interaction error inherent to the standard DFT in describing localized d-electrons with strong correlations, an on-site Hubbard term U-J was added to the open-shell d-electrons, with U = 2 and J = 1 for copper[56]. We tested the solvation effect on *CO adsorption energy using the implicit solvation model implemented in VASPsol[57,58]. The relative permittivity of the media was chosen as 78.4, corresponding to that of water. The results show that the difference of *CO adsorption energy is smaller than 0.1 eV with or without the implicit solvation corrections. The adsorption energy of *CO was calculated with the

$$E_{ads}(M) = E_{total} - E_M - E_{surface} \quad (6)$$

Where $E_{total}$ is the calculated result of the energy of one molecule (*CO) adsorbed on surface, $E_M$ is the energy of an isolated molecule (M), and $E_{surface}$ is the energy of relaxed catalyst.

The computational hydrogen electrode (CHE) model proposed by Nørskov et al. was applied to investigate the free energy profile in CO electrolysis[59]. In CHE method, the relative free energy change is calculated as

$$\Delta G = \mu[product] - \mu[reactant] - 0.5\mu[H_2(g)] + eU \quad (7)$$

Where μ is the chemical potential and U is the applied electrical potential. Therefore, in the step involving proton-electron transfer, $\Delta G(U) = \Delta G_0(U) + neU$, where U is the potential versus the reversible hydrogen electrode, $\Delta G_0$ is the free energy at U = 0 V.

The Gibbs free energy (G) is calculated with Eq. (8).

$$G = E_{Total} + ZPE - TS + \int C_p dT \quad (8)$$

Where $E_{Total}$ is the total electronic energy, ZPE, S and $\int C_p dT$ are the zero-point vibrational energy, entropy, and heat capacity at 298.15 K and 1 atm, respectively. The ZPE, S, and $\int C_p dT$ details of different adsorbates are listed in Supplementary Table 9.

The formation free energy of *COHCO is calculated as

$$\Delta G = G_{*COCOH} - G_{2*CO} - 1/2 G_{H2} \quad (9)$$

Where $G_{*COCOH}$ is the free energy of adsorbed *COCOH on catalyst surface, and $G_{2*CO}$ is the free energy of two adsorbed *CO on catalyst surface, and $G_{H_2}$ is the free energy of $H_2$ molecule. We note that in previous studies, the Gibbs free energy change in the formation of *COCOH from two *CO species was used as indicator of the activity for $C-C$ coupling reaction[18,60]. This provides a convenient method for accessing the catalytic activity differences among various catalysts for $C-C$ coupling. It is also worth noting that the formation energy of *COCOH intermediate shows a strong linear correlation with the activation energy for $C-C$ coupling[61]. Therefore, it is reasonable to predict the activity for $C-C$ coupling reactions on different copper models using the free energy change of *COCOH formation.

## TEA and $CO_2$ emission calculations

The TEA and $CO_2$ emission of CO electrolysis were calculated based on the performance data at an applied current density of 1.0, 3.0, and $4.5\,A\,cm^{-2}$ in the $4\,cm^2$ electrolyzer using previously reported parameters[62–67]. The calculation details were shown in Supplementary Notes 1, 2.

## Data availability

The data that support the findings of this study are available within the paper and the Supplementary Information. Other relevant data are available from the corresponding authors on request. Source data are provided with this paper.

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

## Acknowledgements

This work was supported by the National Key R&D Program of China (2021YFA1501503, D.G.), the National Natural Science Foundation of China (22125205, G.W.; 22372171, D.G.; 22002155, D.G.; 92045302, G.W.), the Fundamental Research Funds for the Central Universities (20720220008, G.W.), the Strategic Priority Research Program of the Chinese Academy of Sciences (XDB0600200, G.W.), the Liao Ning Revitalization Talents Program (XLYC2203178, D.G.), the Dalian Institute of Chemical Physics (DICP I202203, D.G.), the Joint Fund of the Yulin University and the Dalian National Laboratory for Clean Energy (YLU-DNL Fund 2022008, G.W.), the Liaoning Binhai Laboratory (Grant No. LBLF-2023-02, G.W.), and the Photon Science Center for Carbon Neutrality. We thank the staff at the BL11B beamline of the Shanghai Synchrotron Radiation Facility (SSRF) for their technical assistance during the XAS measurements.

## Author contributions

Conceptualization: G.W., X.B., D.G.; Methodology: H.L., D.G., T.L., P.W., M.L., C.W., R.L., J.Y., Z.Z., S.S., Q.F.; Investigation: H.L., P.W.; Visualization: H.L., D.G., T.L.; Funding acquisition: D.G., G.W., X.B.; Supervision: G.W., X.B., D.G.; Writing – original draft: D.G., H.L., T.L.; Writing – review & editing: D.G., H.L., T.L., G.W., X.B.

## Competing interests

The authors declare no competing interests.
