## [Peer Review File · Nature Communications]

REVIEWER COMMENTS

Reviewer #1 (Remarks to the Author):

The paper presented by Li and co-authors is what I think a communication should be. It is dense with data and the team have been able to push the rate and scale of CO reduction electrolysis to a new record. The observed current densities at $>1 \text{ A cm}^{-2}$ are particularly exciting and are an inspiration to other researchers, who have been generally stuck in the $<500 \text{ mA cm}^{-2}$ range. They should also be commended for their first attempts to scale up the catalysts, which is great to see.

The one area where the paper does not present much new is in the explanation of their high activity. The authors' explanation of the high current density being related to the grain boundaries is proven quite adequately, but the novelty of this idea is relatively low. I don't think this is a problem, since the record activity, in my opinion is enough of an advance for publication in Nature Communications.

Nevertheless I do believe some extra experiments and details would be necessary before publication of the manuscript, as listed below:

Can the authors discuss more where their liquid products are collected? Sometimes these are generated at the cathode, but move to the anode. It would help with understanding of the flows in their reactor to have more details here.

The stability tests in general are a little short. I understand that running the stack for long periods may take too many resources for their lab, but good stability tests for the field are now in the 1000 h range at smaller scales. Can the author provide any longer term stability data or is flooding taking over?

Can the authors discuss more the flow of the anolyte to their reactor? Is it recirculating or is it passing through the cell only one time?

Reviewer #2 (Remarks to the Author):

This work reported the electrochemical synthesis of multicarbon products from CO electrolysis using grain boundary-rich Cu nanoparticle catalysts. Using a zero-gap MEA cell, the authors achieved a high C₂₊ partial current density of 4.35 A cm⁻² at a low cell voltage of 2.78 V. Meanwhile, the demonstrated electrocatalytic CO reduction showed better performance – a C₂₊ carbon selectivity of ~100% and a CO conversion of 85% – compared to reported thermocatalytic CO hydrogenation. The authors performed a stack with five 100-cm² MEA at a high total current of 400 A, showing its scale-up potential. I would recommend its publication in Nature Communications after addressing the issues below.

1. The GB density was reduced by annealing in air at high temperature, which also resulted in the formation of CuO. Oxide-derived Cu has been widely reported to be active for CO₂/CO electroreduction. Why is the trend different in this work?

2. The author used a catalyst loading of 2 mg/cm² to prepare the cathode, higher than that in the literature (usually 1 mg/cm² or even less). Is the high current density caused by the high catalyst loading?

3. Stability test was conducted at a very high current density of 1 A/cm² for 150 hours (Figure 1f). How did the authors achieve it? Was the electrode treated with specific surface modification like PTFE? The authors should provide further details about their electrolysis experiments.

4. The techno-economic assessment (TEA) and carbon footprint analysis of CO electrolysis are recommended. They are important parameters for assessing potential industrial applications. The authors calculated them using the performance data at 4.5 A/cm². Operating the cell at such a high current density might be not the economically optimized due to the high cell voltage. The authors are suggested to conduct TEA and carbon footprint analysis using different current densities.

Reviewer #3 (Remarks to the Author):

Wang and co-workers demonstrated an electrochemical and CO₂-free CO conversion alternative route, beyond conventional syngas conversion. CO can be exclusively converted to multicarbon (C₂₊) products via renewable energy-driven electrolysis at industrially-relevant current densities. The authors used a home-made alkaline MEA cell and a grain boundary-rich Cu catalyst (Cu-nc) to achieve an impressive C₂₊ partial current density up to 4.35 A/cm². The productivity of C₂₊ products was further demonstrated by scaling up the CO electrolysis process using a stack with a total electrode area of 500 cm². Moreover, the demonstrated electrocatalytic CO conversion route exhibits remarkable advantages such as C₂₊ selectivity/yield as well as CO₂ emission, compared to state-of-the-art syngas conversion in thermal catalysis. The CO electrolysis performance was positively correlated with GB density. Raman spectroscopic characterization and calculation results

indicated that the GBs of Cu nanoparticles facilitated CO adsorption and C–C coupling, thus resulting in the impressive C₂⁺ production on Cu-nc catalyst with the richest GBs. The data quality is high and this manuscript is well presented. This work would provide important insights into both fundamental understandings and applied studies towards CO electroreduction. Therefore, this manuscript can be recommended for publication in Nature Communications.

Minor revisions below are needed to further improve the manuscript.

1. The scale-up efforts are appreciated. Comparing Figures 1a, 5b, 5e, the scale-up from 4 cm² to 100 cm² and further to a stack led to slight performance decay (increased cell voltage and/or decreased C₂⁺ selectivity), especially for high current densities of 4 and 5 A/cm². The authors should comment on the gradual degradation.

2. In Supplementary Fig. 20, the Cu-550 sample with the lowest GB density also exhibits good C₂⁺ selectivity at 1.5 A/cm², which is still high compared to most studies in the literature. The authors should discuss more on the device and electrode engineering that could contribute to the reported high performance.

3. In the literature, annealing Cu in air followed by electrochemical reduction is known to introduce GBs. In this work, annealing in air indeed reduced the GB density (Figure 2g). Could the authors comment on the difference?

4. Some important relevant references about CO electrolysis (10.1038/s41586-023-05918-8; 10.1126/sciadv.ade3557) are missing and they should be cited and compared.

Reviewer #4 (Remarks to the Author):

The authors described the use of Cu catalysts having a high grain boundary density in electrochemical CORR to generate C₂⁺ products selectively. The experimental results were organized well and reached record-high values in FEs and current densities. It is valuable in demonstrating the potential industrial applications; however, the themes and stories must be more straightforward.

1. The main emphases of this manuscript are twofold: one is the recordable values of catalytic performances and the other is the catalytic activity highly dependent upon the grain boundary density. However, these two themes are not directly related. As shown in Fig. 1e, this work exhibited the best yields and conversion rates among the reported values, not from superior catalyst structures but from advanced device configuration. Then, the authors should describe the details and advantages of their MEA cells to exploit the main reason for such high performances. Unfortunately, their previous paper already utilized the identical device configuration (ref. 13).

2. Depart from the device configuration, the catalysts that they synthesized were not brand-new. Moreover, the catalysts were the precipitates prepared by reducing copper chloride without surfactants; the catalysts were not isolated “nanoparticles” but copper nano-domains connected throughout the materials. The works for the correlation between the activity and GD density were agreeable but not something new.

3. The characterization of GB density was not sufficient. The TEM analysis should be carried out in wider regions, and other technical analyses, such as ED or FFT, should be applied to ensure the difference in GB density.

Hence, the authors should organize this manuscript more reasonably. First, they should describe the excellence of their MEA device configuration using standard catalyst samples, such as Cu foils, OD-Cu, or Cu₂O nanocubes. Then, they should show that their superior catalysts fit the device configuration. The dependency of catalytic activity upon GB density is just an optimizing process for this catalytic device system. The title should also focus on device configuration matching with catalyst design and the recordable values of productivity.

Responses to reviewers comments

Reviewer #1:

The paper presented by Li and co-authors is what I think a communication should be. It is dense with data and the team have been able to push the rate and scale of CO reduction electrolysis to a new record. The observed current densities at $>1 \text{ A cm}^{-2}$ are particularly exciting and are an inspiration to other researchers, who have been generally stuck in the $<500 \text{ mA cm}^{-2}$ range. They should also be commended for their first attempts to scale up the catalysts, which is great to see.

Response: We thank the reviewer for the positive comments on our CO electrolysis performance and scale-up efforts, and the recommendation for publication of our work.

The one area where the paper does not present much new is in the explanation of their high activity. The authors' explanation of the high current density being related to the grain boundaries is proven quite adequately, but the novelty of this idea is relatively low. I don't think this is a problem, since the record activity, in my opinion is enough of an advance for publication in *Nature Communications*.

Response: We thank the reviewer for the comment. Following the similar comments of this reviewer and Reviewers #3 and #4, we have included both GB-rich Cu nanoparticle catalyst and advanced electrolyzer used in our work as main factors that contribute to the high current density.

We have conducted CO electrolysis measurement over commercially available Cu nanoparticle reference catalyst. As shown in Supplementary Fig. 4, the Cu nanoparticle catalyst also shows high C_{2+} FE at a current density up to 3.0 A cm^{-2} , much higher than literature values ($<500 \text{ mA cm}^{-2}$ range). The high electrolyzer performance is ascribed to a hydrophobic GDE structure and the very low Ohmic resistance (as low as $0.13 \text{ } \Omega \cdot \text{cm}^2$, Supplementary Fig. 3) owing to careful optimization in the assembly and operation.

On the other hand, the GB density of our GB-rich Cu nanoparticle powder catalyst reaches $\sim 200 \text{ } \mu\text{m}^{-1}$, 5-fold higher than previously reported carbon-supported Cu nanoparticles (*ACS Cent. Sci.* **2016**, 2, 169). And, the powder catalyst is also very suitable for GDE preparation.

Furthermore, with the GB-rich Cu catalyst incorporated in our MEA electrolyzer, the total current density can further increase to 5.0 A cm^{-2} , resulting in a record C_{2+} partial current density of 4.35 A cm^{-2} .

Therefore, both GB-rich Cu catalyst and the MEA electrolyzer are crucial for our record CO electrolysis performance.

We have added the following text in the revised manuscript.

On Page 4:

“The anode and cathode were fed with 0.5 M KOH solution at a flow rate of 5 mL min^{-1} and dry CO at a flow rate of 80 mL min^{-1} , respectively. The Cu-nc powder catalyst was incorporated into a gas diffusion electrode (GDE) with polytetrafluoroethylene (PTFE) as a binder in the catalyst layer. The hydrophobic and porous GDE structure drastically reduces the diffusion pathway for CO to reach the catalyst, resulting in high current densities^{15,16}. Moreover, through careful optimization in the assembly and operation¹⁷, the MEA electrolyzer used in this work exhibits an Ohmic resistance as low as $0.13 \text{ } \Omega\text{-cm}^2$ (Supplementary Fig. 3), which is very important for reducing cell voltage and increasing full-cell energy efficiency. The high performance of our MEA electrolyzer has been demonstrated using commercially available Cu nanoparticles (Supplementary Fig. 4).”

Supplementary Fig. 3 | Electrochemical impedance spectroscopy (EIS) of MEA electrolyzer with Cu-nc catalyst measured in 0.5 M KOH at open circuit potential after electrolysis at 5.0 A cm^{-2} for 10 min.

Supplementary Fig. 4 | CO electrolysis performance of commercially available Cu nanoparticle catalyst measured in 0.5 M KOH (5.0 mL min⁻¹) under CO feed (80 mL min⁻¹), with a mass loading of 2.0 mg cm⁻² in the electrode.

Nevertheless I do believe some extra experiments and details would be necessary before publication of the manuscript, as listed below:

Can the authors discuss more where their liquid products are collected? Sometimes these are generated at the cathode, but move to the anode. It would help with understanding of the flows in their reactor to have more details here.

Response: We thank the reviewer for the comment. Liquid products were collected from anolyte. In MEA electrolyzer, the liquid products such as ethanol, n-propanol, and acetate generated at the cathode transfer to anode through anion exchange membrane via electromigration under the action of electric field. As shown by our previous work, there are only very little residual liquid products left in the cathode (*Nat. Nanotechnol.* **2023**, *18*, 299). Therefore, we collected liquid products from anolyte.

We have added more details in the revised manuscript.

On Page 15:

“Liquid products collected from anolyte were analyzed by a Bruker AVANCE III 400 MHz nuclear magnetic resonance (NMR) spectrometer. A mixture of anolyte and 1-propanesulfonic acid 3-(trimethylsilyl) sodium salt (DSS, as an internal standard) was used for NMR measurements.”

The stability tests in general are a little short. I understand that running the stack for long periods may take too many resources for their lab, but good stability tests for the field are now in the 1000 h range at smaller scales. Can the author provide any longer term stability data or is flooding taking over?

Response: We thank the reviewer for the important comment. In literature, stability tests are often performed at a relatively low current density (e.g., 200 mA cm⁻²). Performing stability tests at high current density (e.g., 1.0 A cm⁻²) is very important for practical application but quite challenging. By optimizing electrode and electrolyzer structures, we managed to measure the stability of our Cu-nc catalyst at 1.0 A cm⁻² for 150 h in the 4-cm² alkaline MEA electrolyzer, as shown in Fig. 1f.

Stability measurements become more challenging in a larger electrolyzer, since the scale-up of homogeneous gas diffusion electrodes and anion exchange membranes are difficult to control in the lab operation and the heating effect produced by Ohmic resistance is obvious. Nevertheless, in the original manuscript, a stability test at a total current of 100 A (corresponding to a high current density of 1.0 A cm⁻²) for 8 h in the 100-cm² alkaline MEA electrolyzer was shown. Now, we have performed a longer stability test with up to 32 h, as shown in Fig. 5c. The cell voltage is stable at around 2.5 V and the C₂₊ FE maintains above 88% over a course of 32 h.

We have updated the text in the revised manuscript.

On Page 11:

“The stability test conducted at an applied total current of 100 A (1.0 A cm⁻²) shows that the cell voltage is stable at around 2.5 V and the C₂₊ FE maintains above 88% over a course of 32 h (Fig. 5c).”

Fig. 5 | Scale-up demonstration of CO electrolysis. (a) Photograph of 100-cm² GDE. (b) FE and cell voltage as a function of total current over Cu-nc catalyst in the 100-cm² MEA electrolyzer. (c) Stability test at a total current of 100 A in the 100-cm² MEA electrolyzer. (d) Photographs of electrolyzer stack with five 100-cm² MEAs. (e) FE and cell voltage as well as (f) C₂⁺ and C₂H₄ formation rates as a function of total current.

Can the authors discuss more the flow of the anolyte to their reactor? Is it recirculating or is it passing through the cell only one time?

Response: We thank the reviewer for the comment. In order to avoid possible reoxidation of liquid products, the anolyte was not recirculated but passed through the anode only one time. We have added the following text in the revised manuscript.

On Page 15:

“In all above CO electrolysis experiments including stability measurements, the anolyte was not recirculated and fresh anolyte was always used.”

Reviewer #2:

This work reported the electrochemical synthesis of multicarbon products from CO electrolysis using grain boundary-rich Cu nanoparticle catalysts. Using a zero-gap MEA cell, the authors achieved a high C₂⁺ partial current density of 4.35 A cm⁻² at a low cell voltage of 2.78 V.

Meanwhile, the demonstrated electrocatalytic CO reduction showed better performance – a C₂₊ carbon selectivity of ~100% and a CO conversion of 85% – compared to reported thermocatalytic CO hydrogenation. The authors performed a stack with five 100-cm² MEA at a high total current of 400 A, showing its scale-up potential. I would recommend its publication in Nature Communications after addressing the issues below.

Response: We thank the reviewer for the positive comments and the recommendation for publication in Nature Communications.

1. The GB density was reduced by annealing in air at high temperature, which also resulted in the formation of CuO. Oxide-derived Cu has been widely reported to be active for CO₂/CO electroreduction. Why is the trend different in this work?

Response: We thank the reviewer for the comment. In Kanan's pioneering work, oxide-derived Cu was prepared by annealing electropolished Cu foils in air followed by electrochemical reduction during CO₂/CO electroreduction (*J. Am. Chem. Soc.* **2012**, *134*, 7231; *Nature* **2014**, *508*, 504). Higher annealing temperature is used to obtain a thick Cu oxide layer on the Cu foils and thus a high density of GBs. In our work, we synthesized GB-rich Cu nanoparticles by a wet-chemical method and the GB density is as high as ~200 μm⁻¹. As high temperature annealing facilitates recrystallization, here we used annealing at higher temperature to reduce GBs to obtain a series of catalysts with different GB densities

2. The author used a catalyst loading of 2 mg/cm² to prepare the cathode, higher than that in the literature (usually 1 mg/cm² or even less). Is the high current density caused by the high catalyst loading?

Response: We thank the reviewer for the comment. Following the reviewer's comment, we have conducted CO electrolysis measurements over Cu-nc catalysts with lower loadings of 1.0 and 0.5 mg cm⁻², respectively. As shown in Fig. R1, product distribution and current density are just marginally influenced by catalyst loading. The Cu-nc catalyst with a low loading of 0.5 mg cm⁻² can still achieve a high C₂₊ current density of 3.8 A cm⁻², which is slightly lower than that at a high loading of 2.0 mg cm⁻², but is still higher than previous studies with a Cu loading of less than 1.0 mg cm⁻² (e.g., *Nat. Catal.* **2018**, *1*, 748; *Nat. Catal.* **2020**, *3*, 478). Therefore,

we think that the high current density is not attributed to the catalyst loading, but to our GB-rich Cu nanoparticle catalyst and the optimized electrode and electrolyzer configuration.

Fig. R1. FEs and cell voltage as a function of current density over Cu-nc electrodes with catalyst loadings of 0.5 (a), 1.0 (b), and 2.0 (c) mg cm^{-2} .

3. Stability test was conducted at a very high current density of 1 A/cm^2 for 150 hours (Figure 1f). How did the authors achieve it? Was the electrode treated with specific surface modification like PTFE? The authors should provide further details about their electrolysis experiments.

Response: We thank the reviewer for the comment. In our experiments, a cathode gas diffusion electrode (GDE) consists of a gas diffusion layer (GDL, including carbon fiber paper and microporous layer) and a catalyst layer. The catalyst side of the cathode GDE is in contact with the anion exchange membrane, while anolyte transports across the anode and anion exchange membrane to the cathode GDE. The carbon fiber paper side of the cathode GDE is in contact with the cathode flow field and exposed to flowing CO, which diffuses through the pores in the cathode GDE to reach the GB-rich Cu nanoparticle catalyst in the cathode catalyst layer. The addition of PTFE in the cathode GDE including GDL and catalyst layer is to improve the hydrophobic properties, which can maintain the mass transport of the liquid and gas phases in the cathode GDE, respectively, to resist flooding in the pores of the cathode GDE, especially during a stability test at high current density (e.g., 1.0 A cm^{-2}).

We have added the following text in the revised manuscript.

On Page 4:

“The anode and cathode were fed with 0.5 M KOH solution at a flow rate of 5.0 mL min^{-1} and dry CO at a flow rate of 80 mL min^{-1} , respectively. The Cu-nc powder catalyst was incorporated into a gas diffusion electrode (GDE) with polytetrafluoroethylene (PTFE) as a binder in the

catalyst layer. The hydrophobic and porous GDE structure drastically reduces the diffusion pathway for CO to reach the catalyst, resulting in high current densities^{15,16}.”

On Page 13:

“The Cu-nc or Cu-x catalyst and PTFE solution were dispersed in ethanol with a mass ratio of 3:1 to form an ink. The ink was then painted onto the GDL to form a GDE. The catalyst mass loading was $2.0 \pm 0.1 \text{ mg cm}^{-2}$.”

4. The techno-economic assessment (TEA) and carbon footprint analysis of CO electrolysis are recommended. They are important parameters for assessing potential industrial applications. The authors calculated them using the performance data at 4.5 A/cm^2 . Operating the cell at such a high current density might be not the economically optimized due to the high cell voltage. The authors are suggested to conduct TEA and carbon footprint analysis using different current densities.

Response: We thank the reviewer for the positive comment. Following the reviewer’s kind suggestion, based on the performance data reported in our work, we have further performed techno-economic assessment and CO₂ emission analysis for the production of 100 t/day ethylene at an applied current density of 1.0 and 3.0 A cm^{-2} (Supplementary Fig. 39). While the yearly profit decreases with increasing current density, the operation cost and CO₂ emission decrease.

We have added new discussion in the revised manuscript.

On Page 11:

“The CO electrolysis process can be economically viable as demonstrated by techno-economic assessment (TEA) based on the represented performance data at 1.0, 3.0, and 4.5 A cm^{-2} (Supplementary Note 1 and Supplementary Fig. 39), and the CO₂ emission from CO electrolysis can be reduced by up to 82% compared to thermocatalytic CO hydrogenation via Fischer-Tropsch synthesis (Supplementary Note 2 and Supplementary Fig. 39).”

Supplementary Fig. 39 | (a) Yearly profit, (b) operation cost, and (c) CO₂ emission for the production of 100 t/day ethylene based on reported performance data in this work.

Reviewer #3:

Wang and co-workers demonstrated an electrochemical and CO₂-free CO conversion alternative route, beyond conventional syngas conversion. CO can be exclusively converted to multicarbon (C₂₊) products via renewable energy-driven electrolysis at industrially-relevant current densities. The authors used a home-made alkaline MEA cell and a grain boundary-rich Cu catalyst (Cu-nc) to achieve an impressive C₂₊ partial current density up to 4.35 A/cm². The productivity of C₂₊ products was further demonstrated by scaling up the CO electrolysis process using a stack with a total electrode area of 500 cm². Moreover, the demonstrated electrocatalytic CO conversion route exhibits remarkable advantages such as C₂₊ selectivity/yield as well as CO₂ emission, compared to state-of-the-art syngas conversion in thermal catalysis. The CO electrolysis performance was positively correlated with GB density. Raman spectroscopic characterization and calculation results indicated that the GBs of Cu nanoparticles facilitated CO adsorption and C–C coupling, thus resulting in the impressive C₂₊ production on Cu-nc catalyst with the richest GBs. The data quality is high and this manuscript is well presented. This work would provide important insights into both fundamental understandings and applied studies towards CO electroreduction. Therefore, this manuscript can be recommended for publication in Nature Communications.

Response: We thank the reviewer for the positive comments and the recommendation for publication in Nature Communications.

Minor revisions below are needed to further improve the manuscript.

1. The scale-up efforts are appreciated. Comparing Figures 1a, 5b, 5e, the scale-up from 4 cm² to 100 cm² and further to a stack led to slight performance decay (increased cell voltage and/or decreased C₂₊ selectivity), especially for high current densities of 4 and 5 A/cm². The authors should comment on the gradual degradation.

Response: We thank the reviewer for the comment. It is true that the C₂₊ FE slightly decreases and the cell voltage increases when scaling up from 4 cm² to 100 cm² and stack with five 100 cm² MEAs. However, it is quite challenging to achieve the exactly same performance in a much larger electrolyzer (by a factor of 25) and even an electrolyzer stack with complex electric and fluid connection. Moreover, scaling up gas diffusion electrodes and anion exchange membranes is also very difficult to control in the lab operation. The heating effect produced by Ohmic resistance is remarkable at a very high applied current (up to 500 A in this work). Nevertheless, our scale-up efforts indicate the viability of CO electrolysis for the electrochemical synthesis of valuable C₂₊ chemicals.

We have added the following discussion in the revised manuscript.

On Page 11:

“While further efforts should be input in the future to improve the effectiveness and long-term stability of the scale-up process, these scale-up attempts indicate that CO electrolysis is a very promising and practical route for the electrochemical synthesis of valuable C₂₊ chemicals.”

Fig. 5 | Scale-up demonstration of CO electrolysis. (a) Photograph of 100-cm² GDE. (b) FE

and cell voltage as a function of total current over Cu-nc catalyst in the 100-cm² MEA electrolyzer. (c) Stability test at a total current of 100 A in the 100-cm² MEA electrolyzer. (d) Photographs of electrolyzer stack with five 100-cm² MEAs. (e) FE and cell voltage as well as (f) C₂₊ and C₂H₄ formation rates as a function of total current.

2. In Supplementary Fig. 20, the Cu-550 sample with the lowest GB density also exhibits good C₂₊ selectivity at 1.5 A/cm², which is still high compared to most studies in the literature. The authors should discuss more on the device and electrode engineering that could contribute to the reported high performance.

Response: We thank the reviewer for the comment. As our reply to Reviewer #1, we have included both GB-rich Cu nanoparticle catalyst and advanced electrolyzer used in our work as main factors that contribute to the high current density.

We have conducted CO electrolysis measurement over commercially available Cu nanoparticle reference catalyst. As shown in Supplementary Fig. 4, the Cu nanoparticle catalyst also shows high C₂₊ FE at a current density up to 3.0 A cm⁻², much higher than literature values (<500 mA cm⁻² range). The high electrolyzer performance is ascribed to a hydrophobic GDE structure and the very low Ohmic resistance (as low as 0.13 Ω·cm², Supplementary Fig. 3) owing to careful optimization in the assembly and operation.

On the other hand, the GB density of our GB-rich Cu nanoparticle powder catalyst reaches ~200 μm⁻¹, 5-fold higher than previously reported carbon-supported Cu nanoparticles (*ACS Cent. Sci.* **2016**, 2, 169). And, the powder catalyst is also very suitable for GDE preparation. Furthermore, with the GB-rich Cu catalyst incorporated in our MEA electrolyzer, the total current density can further increase to 5.0 A cm⁻², resulting in a record C₂₊ partial current density of 4.35 A cm⁻².

Therefore, both GB-rich Cu catalyst and the MEA electrolyzer are crucial for our record CO electrolysis performance.

We have added the following text in the revised manuscript.

On Page 4:

“The anode and cathode were fed with 0.5 M KOH solution at a flow rate of 5.0 mL min⁻¹ and dry CO at a flow rate of 80 mL min⁻¹, respectively. The Cu-nc powder catalyst was incorporated

into a gas diffusion electrode (GDE) with polytetrafluoroethylene (PTFE) as a binder in the catalyst layer. The hydrophobic and porous GDE structure drastically reduces the diffusion pathway for CO to reach the catalyst, resulting in high current densities^{15,16}. Moreover, through careful optimization in the assembly and operation¹⁷, the MEA electrolyzer used in this work exhibits an Ohmic resistance as low as $0.13 \Omega\text{-cm}^2$ (Supplementary Fig. 3), which is very important for reducing cell voltage and increasing full-cell energy efficiency. The high performance of our MEA electrolyzer has been demonstrated using commercially available Cu nanoparticles (Supplementary Fig. 4).”

Supplementary Fig. 3 | Electrochemical impedance spectroscopy (EIS) of MEA electrolyzer with Cu-nC catalyst measured in 0.5 M KOH at open circuit potential after electrolysis at 5.0 A cm^{-2} for 10 min.

Supplementary Fig. 4 | CO electrolysis performance of commercially available Cu nanoparticle catalyst measured in 0.5 M KOH (5.0 mL min^{-1}) under CO feed (80 mL min^{-1}), with a mass loading of 2.0 mg cm^{-2} in the electrode.

3. In the literature, annealing Cu in air followed by electrochemical reduction is known to introduce GBs. In this work, annealing in air indeed reduced the GB density (Figure 2g). Could the authors comment on the difference?

Response: We thank the reviewer for the comment. In Kanan's pioneering work, oxide-derived Cu was prepared by annealing electropolished Cu foils in air followed by electrochemical reduction during CO₂/CO electroreduction (*J. Am. Chem. Soc.* **2012**, *134*, 7231; *Nature* **2014**, *508*, 504). Higher annealing temperature is used to obtain a thick Cu oxide layer on the Cu foils and thus a high density of GBs. In our work, we synthesized GB-rich Cu nanoparticles by a wet-chemical method and the GB density is as high as ~200 μm⁻¹. As high temperature annealing facilitates recrystallization, here we used annealing at higher temperature to reduce GBs to obtain a series of catalysts with different GB densities

4. Some important relevant references about CO electrolysis (10.1038/s41586-023-05918-8; 10.1126/sciadv.ade3557) are missing and they should be cited and compared.

Response: We thank the reviewer for the comment and we have cited these references (refs. 11,12) in the revised Manuscript.

Reviewer #4:

The authors described the use of Cu catalysts having a high grain boundary density in electrochemical CORR to generate C₂₊ products selectively. The experimental results were organized well and reached record-high values in FEs and current densities. It is valuable in demonstrating the potential industrial applications; however, the themes and stories must be more straightforward.

Response: We thank the reviewer for the positive comments on the manuscript presentation and reported CO electrolysis and scale-up performances in our work. Following the reviewer's important comments and suggestions below, we have conducted further experiments and modified the manuscript thoroughly.

1. The main emphases of this manuscript are twofold: one is the recordable values of catalytic performances and the other is the catalytic activity highly dependent upon the grain boundary density. However, these two themes are not directly related. As shown in Fig. 1e, this work exhibited the best yields and conversion rates among the reported values, not from superior catalyst structures but from advanced device configuration. Then, the authors should describe the details and advantages of their MEA cells to exploit the main reason for such high performances. Unfortunately, their previous paper already utilized the identical device configuration (ref. 13).

Response: We thank the reviewer for the important comment. As our reply to Reviewer #1, we have included both GB-rich Cu nanoparticle catalyst and advanced electrolyzer used in our work as main factors that contribute to the high current density.

We have conducted CO electrolysis measurement over commercially available Cu nanoparticle reference catalyst. As shown in Supplementary Fig. 4, the Cu nanoparticle catalyst also shows high C₂₊ FE at a current density up to 3.0 A cm⁻², much higher than literature values (<500 mA cm⁻² range). The high electrolyzer performance is ascribed to a hydrophobic GDE structure and the very low Ohmic resistance (as low as 0.13 Ω·cm², Supplementary Fig. 3) owing to careful optimization in the assembly and operation.

On the other hand, the GB density of our GB-rich Cu nanoparticle powder catalyst reaches ~200 μm⁻¹, 5-fold higher than previously reported carbon-supported Cu nanoparticles (*ACS Cent. Sci.* **2016**, 2, 169). And, the powder catalyst is also very suitable for GDE preparation. Furthermore, with the GB-rich Cu catalyst incorporated in our MEA electrolyzer the total current density can further increase to 5.0 A cm⁻², resulting in a record C₂₊ partial current density of 4.35 A cm⁻².

Therefore, both GB-rich Cu catalyst and the MEA electrolyzer are crucial for our record CO electrolysis performance.

We have added the following text in the revised manuscript.

On Page 4:

“The anode and cathode were fed with 0.5 M KOH solution at a flow rate of 5.0 mL min⁻¹ and dry CO at a flow rate of 80 mL min⁻¹, respectively. The Cu-nc powder catalyst was incorporated into a gas diffusion electrode (GDE) with polytetrafluoroethylene (PTFE) as a binder in the

catalyst layer. The hydrophobic and porous GDE structure drastically reduces the diffusion pathway for CO to reach the catalyst, resulting in high current densities^{15,16}. Moreover, through careful optimization in the assembly and operation¹⁷, the MEA electrolyzer used in this work exhibits an Ohmic resistance as low as $0.13 \Omega\text{-cm}^2$ (Supplementary Fig. 3), which is very important for reducing cell voltage and increasing full-cell energy efficiency. The high performance of our MEA electrolyzer has been demonstrated using commercially available Cu nanoparticles (Supplementary Fig. 4).”

Supplementary Fig. 3 | Electrochemical impedance spectroscopy (EIS) of MEA electrolyzer with Cu-nc catalyst measured in 0.5 M KOH at open circuit potential after electrolysis at 5.0 A cm^{-2} for 10 min.

Supplementary Fig. 4 | CO electrolysis performance of commercially available Cu nanoparticle catalyst measured in 0.5 M KOH (5.0 mL min^{-1}) under CO feed (80 mL min^{-1}), with a mass loading of 2.0 mg cm^{-2} in the electrode.

2. Depart from the device configuration, the catalysts that they synthesized were not brand-new. Moreover, the catalysts were the precipitates prepared by reducing copper chloride without surfactants; the catalysts were not isolated “nanoparticles” but copper nano-domains connected throughout the materials. The works for the correlation between the activity and GD density were agreeable but not something new.

Response: We thank the reviewer for the comment. In Kanan’s pioneering work, oxide-derived Cu was prepared by annealing electropolished Cu foils in air followed by electrochemical reduction during CO₂/CO electroreduction (*J. Am. Chem. Soc.* **2012**, *134*, 7231; *Nature* **2014**, *508*, 504). Higher annealing temperature is used to obtain a thick Cu oxide layer on the Cu foils and thus a high density of GBs. In our work, we synthesized GB-rich Cu nanoparticles by a wet-chemical method and the GB density is as high as $\sim 200 \mu\text{m}^{-1}$, 5-fold higher than previously reported carbon-supported Cu nanoparticles (*ACS Cent. Sci.* **2016**, *2*, 169).

As the reviewer mentioned, the GB-rich Cu nanoparticle catalyst is highly porous with interconnected nanocrystalline networks comprised of nanosized Cu domains, as shown in SEM and TEM images (Fig. 2a, Supplementary Figs. 8,9). This indeed results in the very high GB density of our Cu catalyst. While the correlation between the activity and GD density was reported, the CO electrolysis performance was low. In our work, combining our Cu catalyst with a very high GB density and the advanced MEA electrolyzer, we have managed to achieve a record C₂₊ partial current density as high as 4.35 A cm⁻².

We have added the following text in the revised manuscript.

On Page 6:

“Such a high GB density is ascribed to the interconnected networks comprised of nanosized Cu domains (Fig. 2a, Supplementary Figs. 8,9).”

Fig. 2 | Correlation between C_{2+} production and GB density. (a) SEM image of Cu-nc catalyst. HRTEM images of (b) Cu-nc, (c) Cu-nc after CO electrolysis, (d) Cu-350, (e) Cu-450, and (f) Cu-550 catalysts. (g) GB densities of Cu-nc, Cu-350, Cu-450, and Cu-550 catalysts. The error bars represent standard error of the mean and are made based on fully separate and identical measurements. (h) Correlations between ECSA-normalized C_{2+} /ethylene/acetate partial current densities and GB density at a cell voltage of 2.6 V.

Supplementary Fig. 8 | SEM images of (a) Cu-350, (b) Cu-450, and (c) Cu-550 catalysts in their as-prepared state.

Supplementary Fig. 9 | TEM images of (a) Cu-nc, (b) Cu-350, (c) Cu-450, and (d) Cu-550 catalysts in their as-prepared state.

3. The characterization of GB density was not sufficient. The TEM analysis should be carried out in wider regions, and other technical analyses, such as ED or FFT, should be applied to ensure the difference in GB density.

Response: We thank the reviewer for the useful comment. The GB density of each sample was quantified by analyzing ten typical HRTEM images. Now we have added FFT patterns of grains present on the surface of each sample.

We have updated the text in the revised manuscript.

On Page 6:

“While the porous structure remains over the Cu-x catalysts after annealing in air, the number of GBs drastically decreases (Fig. 2d–f, Supplementary Figs. 10,12–14). The densities of GBs present in the Cu-nc and Cu-x catalysts were quantified by analyzing ten typical HRTEM images for each sample (Supplementary Figs. 11–18).”

Supplementary Fig. 10 Representative HRTEM images and the corresponding of FFT patterns acquired from each grain over as-prepared (a) Cu-nc, (b) Cu-350, (c) Cu-450, and (d) Cu-550 catalysts.

Hence, the authors should organize this manuscript more reasonably. First, they should describe the excellence of their MEA device configuration using standard catalyst samples, such as Cu foils, OD-Cu, or Cu₂O nanocubes. Then, they should show that their superior catalysts fit the device configuration. The dependency of catalytic activity upon GB density is just an optimizing process for this catalytic device system. The title should also focus on device configuration matching with catalyst design and the recordable values of productivity.

Response: We thank the reviewer for these important and useful comments. Following these comments, we have thoroughly modified our manuscript, highlighting both catalyst structure and device configuration.

As our reply to Comment 1, we have performed CO electrolysis measurements over a commercially available Cu nanoparticle reference catalyst sample using the same MEA electrolyzer. As shown in Supplementary Fig. 4, the Cu nanoparticle catalyst also shows high C₂₊ FE at a current density up to 3.0 A cm⁻², much higher than literature values (<500 mA cm⁻² range). The high electrolyzer performance is ascribed to a hydrophobic GDE structure and the very low Ohmic resistance (as low as 0.13 Ω·cm², Supplementary Fig. 3) owing to careful optimization in the assembly and operation.

On the other hand, the GB density of our GB-rich Cu nanoparticle powder catalyst reaches $\sim 200 \mu\text{m}^{-1}$, 5-fold higher than previously reported carbon-supported Cu nanoparticles (*ACS Cent. Sci.* **2016**, 2, 169). Different the earliest GB-rich oxide-derived Cu (OD-Cu) foil samples (*J. Am. Chem. Soc.* **2012**, 134, 7231; *Nature* **2014**, 508, 504), our Cu catalyst is a powder sample which is very suitable for GDE preparation and being incorporated into MEA electrolyzers. Thus, the GB-rich Cu catalyst matches well with our MEA electrolyzer.

Therefore, both GB-rich Cu catalyst and the MEA electrolyzer are crucial for our record CO electrolysis performance. Combining our Cu catalyst with a very high GB density and the advanced MEA electrolyzer, we have managed to achieve a record C_{2+} partial current density as high as 4.35 A cm^{-2} .

Following the reviewer's suggestion, we have also changed the title to "*CO electrolysis to multicarbon products over grain boundary-rich Cu nanoparticles in membrane electrode assembly electrolyzers*" in the revised manuscript.

We have also added the new discussion in the revised manuscript.

On Page 4:

"The anode and cathode were fed with 0.5 M KOH solution at a flow rate of 5.0 mL min^{-1} and dry CO at a flow rate of 80 mL min^{-1} , respectively. The Cu-nc powder catalyst was incorporated into a gas diffusion electrode (GDE) with polytetrafluoroethylene (PTFE) as a binder in the catalyst layer. The hydrophobic and porous GDE structure drastically reduces the diffusion pathway for CO to reach the catalyst, resulting in high current densities^{15,16}. Moreover, through careful optimization in the assembly and operation¹⁷, the MEA electrolyzer used in this work exhibits an Ohmic resistance as low as $0.13 \Omega\text{-cm}^2$ (Supplementary Fig. 3), which is very important for reducing cell voltage and increasing full-cell energy efficiency. The high performance of our MEA electrolyzer has been demonstrated using commercially available Cu nanoparticles (Supplementary Fig. 4)."

On Page 11:

"We present an impressive CO electrolysis performance with a record C_{2+} partial current density as high as 4.35 A cm^{-2} at a low cell voltage of 2.78 V in a home-made alkaline MEA electrolyzer."

REVIEWERS' COMMENTS

Reviewer #1 (Remarks to the Author):

The authors have adequately addressed all of my comments, I would be happy to see the paper published. I believe the authors have done a great job and wish them all the best.

Reviewer #2 (Remarks to the Author):

The authors have addressed my comments. The current version is suitable now for the publication in Nature Communications.

Reviewer #3 (Remarks to the Author):

The authors have addressed my comments very well. The manuscript could be published in the Nature Communications as it is.

Reviewer #4 (Remarks to the Author):

The authors tried to follow up the reviewers' comments properly. Although the record-high activities are highly dominated by the device configuration (as shown in the experiments using commercially available Cu powders), their achievements of highly active and stable catalytic systems are valuable enough to approach practical applications. Hence, this manuscript is now publishable without further revisions.